# Interaction of the p.Q141K Variant of the *ABCG2* Gene with Clinical Data and Cytokine Levels in Primary Hyperuricemia and Gout

**DOI:** 10.3390/jcm8111965

**Published:** 2019-11-14

**Authors:** Veronika Horváthová, Jana Bohatá, Markéta Pavlíková, Kateřina Pavelcová, Karel Pavelka, Ladislav Šenolt, Blanka Stibůrková

**Affiliations:** 1Institute of Rheumatology, 128 50 Prague, Czech Republic; veronika@horvath.cz (V.H.); bohata@revma.cz (J.B.); pavelcova@revma.cz (K.P.); pavelka@revma.cz (K.P.); senolt@revma.cz (L.Š.); 2Faculty of Science, Charles University, 128 00 Prague, Czech Republic; 3Department of Rheumatology, First Faculty of Medicine, Charles University, 128 50 Prague, Czech Republic; 4Department of Probability and Mathematical Statistics, Faculty of Mathematics and Physics, Charles University, 186 75 Prague, Czech Republic; marketa@ucw.cz; 5Department of Pediatrics and Adolescent Medicine, First Faculty of Medicine, Charles University, General University Hospital in Prague, 120 00 Prague, Czech Republic

**Keywords:** p.Q141K, *ABCG2*, cytokines, gout, hyperuricemia, acute gouty arthritis

## Abstract

Gout is an inflammatory arthritis influenced by environmental risk factors and genetic variants. The common dysfunctional p.Q141K allele of the *ABCG2* gene affects gout development. We sought after the possible association between the p.Q141K variant and gout risk factors, biochemical, and clinical determinants in hyperuricemic, gouty, and acute gouty arthritis cohorts. Further, we studied the correlation of p.Q141K allele and levels of pro-/anti-inflammatory cytokines. Coding regions of the *ABCG2* gene were analyzed in 70 primary hyperuricemic, 182 gout patients, and 132 normouricemic individuals. Their genotypes were compared with demographic and clinical parameters. Plasma levels of 27 cytokines were determined using a human multiplex cytokine assay. The p.Q141K variant was observed in younger hyperuricemic/gout individuals (*p* = 0.0003), which was associated with earlier disease onset (*p* = 0.004), trend toward lower BMI (*p* = 0.056), and C-reactive protein (CRP, *p* = 0.007) but a higher glomerular filtration rate (GFR, *p* = 0.035). Levels of 19 cytokines were higher, mainly in patients with acute gouty arthritis (*p* < 0.001), irrespective of the presence of the p.Q141K variant. The p.Q141K variant influences the age of onset of primary hyperuricemia or gout and other disease-linked risk factors and symptoms. There was no association with cytokine levels in the circulation.

## 1. Introduction

Gout is the most common type of inflammatory arthritis in adults that develops as a consequence of elevated urate levels. Gout has four phases: asymptomatic hyperuricemia, acute gouty arthritis, intercritical gout, and chronic tophaceous gout [1]. The attacks are caused by an inflammatory response to monosodium urate (MSU) crystals that deposit in joints, tendons, and surrounding tissues. Before the development of gout, patients show asymptomatic hyperuricemia (elevated levels of serum uric acid >420 µmol/L for men and >360 µmol/L for women). Not all individuals with hyperuricemia develop symptomatic gout, but the risk increases in proportion to the elevation of urate in circulation. Heritability of serum urate levels and gout in Europeans has been estimated to be approximately 30% [2]. 

The mechanism of gouty inflammation is coupled with the formation and activation of the NOD-like receptor P3 (NLRP3) inflammasome with subsequent production of pro-inflammatory cytokines. MSU crystals by themselves are not responsible for the induction of expression and assembly of inflammasome components. The first signal involves priming monocyte-derived macrophages, which then start binding ligands (S100A8 and S100A9 proteins, long-chain free fatty acids) to Toll-like receptors (TLR) [3]. The second signal, triggered by MSU crystals, leads to activation of multiprotein intracellular NLRP3 inflammasomes, which contain pro-caspase 1 [4]. Next, active caspase 1 cleaves precursors of interleukins 1 beta and 18 (pro-IL-1β and pro-IL-18) to generate the active forms (IL-1β and IL-18) [5,6]. Secretion of IL-1β leads to recruitment of neutrophils to the site of inflammation, production of additional pro-inflammatory cytokines, and bone/cartilage degradation [3,7]. IL-1β and IL-18 induced other pro-inflammatory cytokines IL-6, IL-8, IL-17, and tumor necrosis factor alpha (TNFα), which synergistically potentiate gouty inflammation [8,9,10].

Risk of gout development is conditioned not only by hyperuricemia but also by gender, weight, age, environmental and genetic factors, and their interactions. The main cause of hyperuricemia is a defect in renal excretion of urate. Today 10 genes for the main urate transporters are known, including ATP-binding cassette subfamily G member 2 (*ABCG2*), which has the greatest influence on urate excretion. The ABCG2 protein is a homodimeric membrane transporter with functions in a variety of tissues, including xenobiotic transport. Many population-specific variants of the *ABCG2* gene have been found e.g., the common non-synonymous variants p.Q126X (rs72552713), p.V12M (rs2231137), and p.Q141K (rs2231142) [11,12]. The p.Q126X variant is specific to East Asian populations and has a well-known genetic impact on gout induction. It decreases ABCG2 urate transport by 96%. A second common variant in Caucasians is p.V12M, which showed a protective role in gout development and has no influence on urate transport in human embryonic kidney derived cells [13,14]. The missense p.Q141K variant (minor frequency allele (MAF), present in the Central European Caucasian population at around 9.4%) is localized in the nucleotide-binding domain and leads to decreased binding of adenosine triphosphate (ATP) and reduces transport function by almost 50% [14,15,16]. Carriers of p.Q141K tend to have a higher chance of developing an earlier gout onset and have an increased risk of a poor response to allopurinol [12,17,18,19,20]. Moreover, p.Q141K may also influence progression from asymptomatic hyperuricemia to gout by promoting the immune response to MSU crystals. This suggests a hypothetic molecular pathway that connects non-functional variants of urate transporters and the innate immune response [1]. In previous studies, pro-inflammatory cytokines affected *ABCG2* gene expression. Its expression could be upregulated or downregulated by IL-1β and TNFα treatment, either alone or in combination with other biologically active molecules such as estrogen [21,22,23]. 

In our previous studies using an *ABCG2* analysis of Czech patients with hyperuricemic/gout, we found that both the rare and common p.Q141K variants are independently associated with hyperuricemia and gout [12,13]. Moreover, we found a significantly higher frequency of the minor allele p.Q141K variant in pediatric-onset patients when compared to adult-onset patients [24]. Our latest published study demonstrated the effects of rare variants on the expression, cellular localization, and function of ABCG2 [25]. 

The aim of this study was to analyze (1) the association of the p.Q141K variant with known risk factors of gout (e.g., age, BMI, etc.), (2) the possible link between the risk of the p.Q141K allele and levels of biochemical parameters connected with gout (CRP, estimated glomerular filtration rate calculated using the Modification of Diet in Renal Disease (eGFR-MDRD)), and finally (3) the relationship between the p.Q141K allele and pro-/anti-inflammatory cytokines using human multiplex cytokine assay in patients with primary hyperuricemia, gout, acute gouty arthritis, and normouricemic individuals. Relationships between the p.Q141K variant and cytokine plasma levels were addressed for the first time.

## 2. Experimental Section

### 2.1. Cohort and Subcohorts

The study cohort comprised 182 patients with primary gout (166 males, 16 females; median age = 54 years), including 17 patients (23 samples) having a gout attack, and 70 patients with asymptomatic hyperuricemia (50 males, 20 females; median age = 38 years) from the biobank of the Institute of Rheumatology, Prague, the Czech Republic. Gout patients met the 1977 American Rheumatism Association preliminary classification criteria [26]. Primary hyperuricemic patients were classified as serum uric acid (SUA) >420 µmol/L for men and SUA > 360 µmol/L for women on two measurements. Two repetitive measurements of purine metabolites (xanthine, hypoxanthine, and oxypurinol), serum uric acid (SUA), and biochemical parameters were performed to exclude the impact of purine metabolic disorders in all patients. The first xanthine, hypoxanthine, and oxypurinol metabolites, SUA, and biochemical determination were performed during allopurinol/febuxostat treatment. The second measurement was done before the start of treatment or 72 h after a temporary interruption of SUA lowering therapy. Patients were compared to 132 normouricemic individuals from the general population, with no history of primary hyperuricemia, gout, or autoimmune disease (54 males, 78 females; median age = 41 years). All patients and normouricemic subjects were examined for the functional p.Q141K variant of the *ABCG2* gene, as published previously [13]. All demographic (age of disease onset and age of examination/sampling, sex, body mass index (BMI)), biochemical (SUA with/off treatment, fractional excretion of uric acid (FE-UA), estimated glomerular filtration rate calculated using the Modification of Diet in Renal Disease formula (eGFR-MDRD), serum creatinine, and maximal C-reactive protein (CRP)), genetic (familial occurrence, presence of the wild-type p.Q141K variant (GG), heterozygotic (GT), and homozygotic (TT) form and its MAF) and presence and type of medical treatment characteristics of patients and normouricemic individuals are stated in Table 1. This study was approved by the local ethics committee, and all patients and normouricemic individuals signed informed consent. 

### 2.2. Cytokine Detection

In total, 27 cytokines (IL-1β, IL-1ra, IL-2, IL-4, IL-5, IL-6, IL-7, IL-8, IL-9, IL-10, IL-12, IL-13, IL-15, IL-17, IP-10, MCP-1, FGF-basic, eotaxin, G-CSF, GM-CSF, IFN-γ, MIP-1α, MIP-1β, PDGF, RANTES, TNF-α, and VEGF) were determined in 149 selected plasma samples from 132 (out of 182) gout patients, in 23 samples from 17 gout patients during an inflammatory attack, in 44 (out of 70) hyperuricemic patients, and in 132 normouricemic subjects. Samples were tested using commercial Bio-Plex Pro TM Human Cytokine 27-plex Assay kits (Bio-Rad, California, USA) and a Bio-Plex 200 system (Bio-Rad, California, USA). Plasma samples were prepared using a standard protocol and aliquoted to storage at −80 °C until analyzed. 

### 2.3. Statistical Analysis

Data were summarized as means with standard deviations (SD) and/or medians with interquartile ranges (IQR) where appropriate. Continuous subject characteristics between groups were compared using the Kruskal–Wallis ANOVA, and categorical characteristics were compared using the Fisher exact test. Associations between characteristics of the diagnostic group (i.e., with the normouricemic, hyperuricemic, or gouty status) and the p.Q141K allele dose were modeled using linear regression with interactions; creatinine and CRP values were log-transformed for a better fit. Associations between cytokine measurements and diagnostic groups and the p.Q141K allele dose were modeled using linear mixed models with group and allele dose as fixed effects and subject ID as the random factor since some individuals were measured multiple times. All cytokine measurements, except for eotaxin, were log-transformed for a better fit. The Benjamini–Hochberg multiple comparisons correction was used where appropriate. All analyses were performed using statistical language and environment *R*, version 3.4.4.

## 3. Results

### 3.1. Analysis of the p.Q141K Variant in Normouricemic and Patient Cohort

First, we analyzed the presence of the p.Q141K allele in the *ABCG2* gene in our study cohort. From 132 normouricemic subjects with an MAF = 7.1%, 15 were heterozygote, and one was homozygote for p.Q141K variant. In the hyperuricemic cohort with an MAF = 21.3%, 19 were heterozygous, and five were homozygous for the p.Q141K allele. Finally, in gout patients with an MAF = 25%, 68 patients were GT heterozygotes and 11 were TT homozygotes. Gout and hyperuricemic patients had significantly more heterozygous and homozygous variants and hence, a significantly higher p.Q141K frequency than normouricemic subjects (*p* < 0.0001) (Table 1).

### 3.2. Analysis of the p.Q141K Variant and Its Relationship with Risk Factors for Gout

Age, BMI, CRP, and GFR are known risk factors for gout. For each of these, we found expected differences between the diagnostic groups but their association with a diagnostic status varied depending on the p.Q141K allele dose.

Risk of gout and hyperuricemia increases proportionally with age. In our cohort, patients with hyperuricemia or gout were older than normouricemic individuals (*p* < 0.0001, Figure 1A) during the time of the study. However, hyperuricemic and gout patients homozygous for p.Q141K were significantly younger than wild-types and heterozygotes (*p* = 0.0003). Furthermore, hyperuricemia and gout manifested earlier in p.Q141K homozygotes (*p* = 0.004, Figure 1B). 

Elevated BMI increases the probability of metabolic syndrome as well as the probability of hyperuricemia and gout. This is in line with our data: Patients with hyperuricemia or gout had a higher BMI than normouricemic patients (*p* < 0.0001). In contrast, homozygotes for the p.Q141K variant tended to have lower BMIs (*p* = 0.056, Figure 1C). The homozygotes with lower BMIs were mostly very similar to those with young-age hyperuricemia onset.

Similarly, to age and BMI, CRP levels were higher in hyperuricemic and gouty patients than in normouricemic individuals (*p* < 0.0001). But significantly lower levels of CRP were measured in p.Q141K homozygous hyperuricemic and gout patients (*p* = 0.007, Figure 2A). 

Although GFR should be decreased in patients with hyperuricemia and gout because of worsened renal function, heterozygous and mainly homozygous carriers of the p.Q141K variant had significantly higher GFRs (*p* = 0.035, Figure 2B, data for normouricemic subjects were not available).

### 3.3. Analysis of the p.Q141K Variant in Association with Cytokine Plasma Levels

To determine the possible differences between normouricemic, hyperuricemic, gout, and acute gout patients on immunological levels, we determined the cytokine plasma profiles of our cohort. Then, we analyzed the possible influence of the p.Q141K variant on cytokine production between patient groups. 

We found that levels of IL-1β, IL-1ra, IL-4, IL-6, IL-7, IL-8, IL-9, IL-13, IL-17, fibroblast growth factor-basic (FGF-basic), interferon gamma (IFNγ), and TNFα were significantly increased in patients with acute gouty arthritis compared to other groups (multiple correction *p*-values ranging from *p* < 0.05 to *p* < 0.001, see Figure 3). On the other hand, eotaxin, monocyte chemoattractant protein-1 (MCP-1), and regulated on activation, normal T cell expressed and secreted (RANTES) were decreased in all patient groups compared to normouricemic individuals (multiple correction *p*-values ranging *p* < 0.001, *p* = 0.014, and *p* < 0.001, respectively). IL-5, IL-10, and IL-12 levels gradually decreased from control, subsequently hypouricemia, and gout to gout attack group. Plasma levels of IL-2, IL-15, granulocyte-colony stimulating factor (G-CSF), granulocyte-macrophage colony stimulating factor (GM-CSF), interferon gamma-induced protein-10 (IP-10), macrophage inflammatory protein-1 beta (MIP-1β), platelet derived growth factor (PDGF), and vascular endothelial growth factor (VEGF) were comparable among all groups (Figure 3, the *p*-values shown are after the Benjamini–Hochberg correction). We did not find any impact of p.Q141K on the plasma cytokine levels of patients or normouricemic subjects.

## 4. Discussion

The ABCG2 protein is a membrane-associated transporter with a wide range of functions including excretion of urate. The *ABCG2* gene is expressed in the intestine, liver, kidney, and several other tissues that have a barrier function. The renal excretion of urate accounts for approximately two-thirds of total urate excretion while intestinal excretion accounts for the rest [27]. Dysfunction in this protein is particularly caused by the p.Q141K variant, which occurs at a high frequency in the Caucasian population. The variant leads to hyperuricemia and gout. Homozygotes for p.Q141K have 53% reduced excretion of urate compared to wild types [28]. Hypothetically, this allele also promotes progression from hyperuricemia to gout by stimulating the immune response to MSU crystals [1]. Therefore, we looked for a possible relationship between the number of p.Q141K alleles (i.e., wild type, heterozygous, and homozygous) and risk factors (age, BMI) or related determinants (CRP, GFR), and plasma levels of pro/anti-inflammatory cytokines.

Risk of gout increases with age, which is in line with our results where patients with hyperuricemia or gout were older than normouricemic individuals [13,29]. However, hyperuricemic and gout patients homozygous for p.Q141K were significantly younger than wild-types and heterozygotes. They also had an earlier disease onset. These findings are supported by the fact that non-functional ABCG2 transporters have a greater impact on disease progression (PAR0% = 29.2%) than risk factors such as age (PAR% = 5.74%), obesity (PAR% = 18.7%), and alcohol consumption (PAR% = 15.4%) [29].

Another risk factor for hyperuricemia and gout as well as for metabolic syndrome is increased BMI [30]. This corresponds to our data showing that patients with hyperuricemia or gout had higher BMIs than normouricemic individuals. In contrast with previously mentioned data, homozygotes for the p.Q141K variant tended to have lower BMIs. These patients were very similar to those with young-age onset. This finding suggests that the beginning of the disease was mainly influenced by the presence of p.Q141K rather than risk factors such as BMI.

In addition to the fact that high CRP levels indicate infection- or non-infection-based inflammation, serum CRP levels positively correlate with serum urate levels and can be used as a gout determinant [31]. Moreover, CRP levels increase with age and the amount of adipose tissue [32,33]. In this study, we found higher CRP levels in hyperuricemic and gouty patients than in normouricemic controls. However, significantly lower levels of CRP were found in p.Q141K homozygotes. These results show a similar trend as BMI, which could be explained by the younger age of the homozygous p.Q141K allele carriers.

GFR defines kidneys function and deteriorates in patients with hyperuricemia and gout due to worsened renal function [34]. In our study, heterozygous and mainly homozygous carriers of the p.Q141K variant had significantly higher eGFR-MDRD than patients with the wild-type variant. This finding could be explained by the gradually decreasing age of disease onset between wild-type, heterozygous, and homozygous carriers of the p.Q141K variant, although, mainly in hyperuricemic patients. Although eGFR-MDRD was not measured in normouricemic controls, all patients reached eGFR-MDRD reference levels; thus, we did not observe deteriorated renal function in our patient cohort. Another explanation for the high eGRF-MDRD could be the low mean age (53.6 years) of our study cohort compared to the mean age (61.9 years) of the gout patients described in the literature [35]. Our patients were younger, and they did not have impaired kidney function. The higher eGFR-MDRD in homozygotes might not be representative because of the low number of patients (five with hyperuricemia and seven with gout). Further experiments on a larger cohort are necessary to confirm or disprove this particular finding.

Another aim of our study was to examine the differences in cytokine levels between normouricemic, hyperuricemic, gouty, and acute gouty arthritis patients in the presence of the p.Q141K variant of the *ABCG2* gene. Expression of the *ABCG2* gene is impacted by pro-inflammatory cytokines and signaling cascades. One of the major transcription factors of pro-inflammatory pathways, nuclear factor kappa B (NF-κB), is able to bind to the promoter of the *ABCG2* gene and upregulate it, with or without the influence of an estrogen receptor [23,36]. On the other hand, the *ABCG2* gene alone is capable of inhibiting NF-κB activation, which leads to the downregulation of IL-8 [37]. Pro-inflammatory cytokines IL-1β and TNFα can have an impact on *ABCG2* expression at the mRNA as well as the protein level, depending on the kind of cell line. Further, the ABCG2 protein is seen to be downregulated in colon biopsies from patients with active inflammatory bowel disease [38,39,40]. The presence of the p.Q141K variant in the ABCG2 protein leads to misfolding and decreased cell surface expression [17]. These data led us to study the impact of p.Q141K on cytokine levels. Unfortunately, we did not find any influence of this variant on cytokine protein expression in the plasma of patients from different groups or in normouricemic individuals. The likely reason our hypothesis failed was the relatively small number of patients included in the study. Further investigations using larger cohorts are needed.

Nonetheless, we found higher plasma levels of IL-1β, IL-1ra, IL-4, IL-6, IL-8, IL-9, IL-13, IL-17, FGF-basic, IFNγ, and TNFα in acute gouty arthritis patients. All these cytokines are involved in interconnected pro-/anti-inflammatory cascades, which suggest an active immune response in patients with acute gouty arthritis. On the other hand, IL-5, IL-10, and eotaxin were lower in patients with acute gout inflammation, which can be explained by their role in inhibiting inflammation (IL-10) or eosinophilic inflammation (IL-5, eotaxin) [41]. Our results regarding IL-10 levels are supported by similar findings of circulating IL-10 in a Netherlands gouty arthritis study cohort and in New Zealand patients with intercritical gout study [42]. Estevez-Garcia and colleagues measured serum levels of IL-5 among groups of normouricemic controls, hyperuricemic, and gouty patients [43] and found, in contrast to our results, IL-5 levels were the same among groups, which may be explained by IL-5 determination in serum vs. plasma. Plasma is a more sensitive matrix for detecting levels of low-abundance cytokines [44]. Eotaxin was described to be increased in obese patients with metabolic syndrome (mean BMI 38.9 ± 6.3) compared to lean subjects (mean BMI 22.4 ± 2.5) and decreased after their weight-loss [45]. This suggests that eotaxin should have been higher in our patients than in normouricemic individuals; however, our patients were not extremely obese. MCP-1 and RANTES were also lower in acute gouty arthritis patients compare to normouricemic controls in our study. However, increased serum levels of MCP-1 have been observed in hyperuricemic, gouty, and acute gouty arthritis patients, and MCP-1 has been positively correlated with elevated serum levels of uric acid [46]. The contradictory results could be explained by measurement of cytokines in plasma samples in our study. To date no association of RANTES plasma/serum levels with gout has been published; however, it is known that urate stimulates mRNA and protein expression of RANTES and MCP-1 in mice tubular epithelial cells [47]. On the other hand, synovial fluid levels of RANTES were found to be significantly lower in patients with acute gout than in patients with acute rheumatoid arthritis [48].

One of the limitations of our study was the relatively small number of p.Q141K homozygotes with hyperuricemia and gout and also a small cohort size of those with acute gouty arthritis. A larger group for p.Q141K homozygotes and acute gout arthritic patients is necessary to confirm the lower age of onset, BMI, CRP, and higher eGFR-MDRD, and cytokine levels between groups. Our results reflect the fact that our study cohort was purposefully compiled from patients with primary hyperuricemia and gout who are younger (mean age = 53.6 years) than the common gouty population (mean age = 61.9 years) and have a positive family correlation [35]. This characteristic can be considered to be one of the strengths of the study because the cohort was well-defined and excluded those with secondary hyperuricemia and gout. Our study also used explicit definitions of medication and comorbidities for the study cohorts. Finally, according to available data, and for the first time, our study examined a possible link between the p.Q141K variant and plasma cytokine levels.

## 5. Conclusions

In conclusion, we found that the p.Q141K variant of the *ABCG2* gene impacts the age of hyperuricemia and gout onset, levels of BMI and CRP, and renal function. This variant has no effect on plasma cytokine levels of hyperuricemic and gouty patients. However, we did find higher cytokine levels, but mainly in patients with acute gouty arthritis.

## Figures and Tables

**Figure 1 jcm-08-01965-f001:**
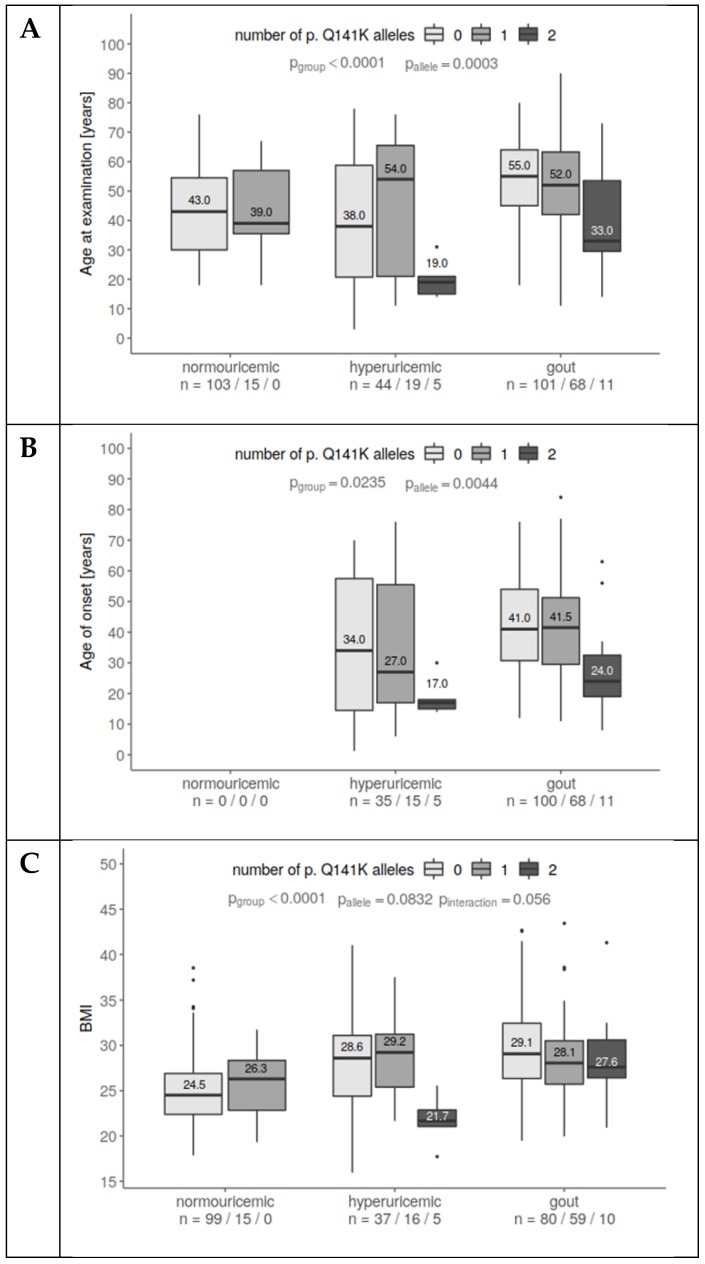
The association of p.Q141K alleles with age at examination, age of onset, and BMI. (**A**) Gout patients were older than hyperuricemic and normouricemic subjects. Homozygotes for p.Q141K were younger than wild type/heterozygotes. Interactions suggest different associations for normouricemic subjects. (**B**) There are no substantial differences in the age of onset between hyperuricemia and gout individuals. p.Q141K homozygotes tend to have much earlier disease onset than others. (**C**) Gout and hyperuricemic patients had higher BMIs than controls. Homozygous patients were slightly leaner than others. A statistically significant interaction suggests different associations for p.Q141K among groups. This may have been influenced by the pediatric hyperuricemia subset.

**Figure 2 jcm-08-01965-f002:**
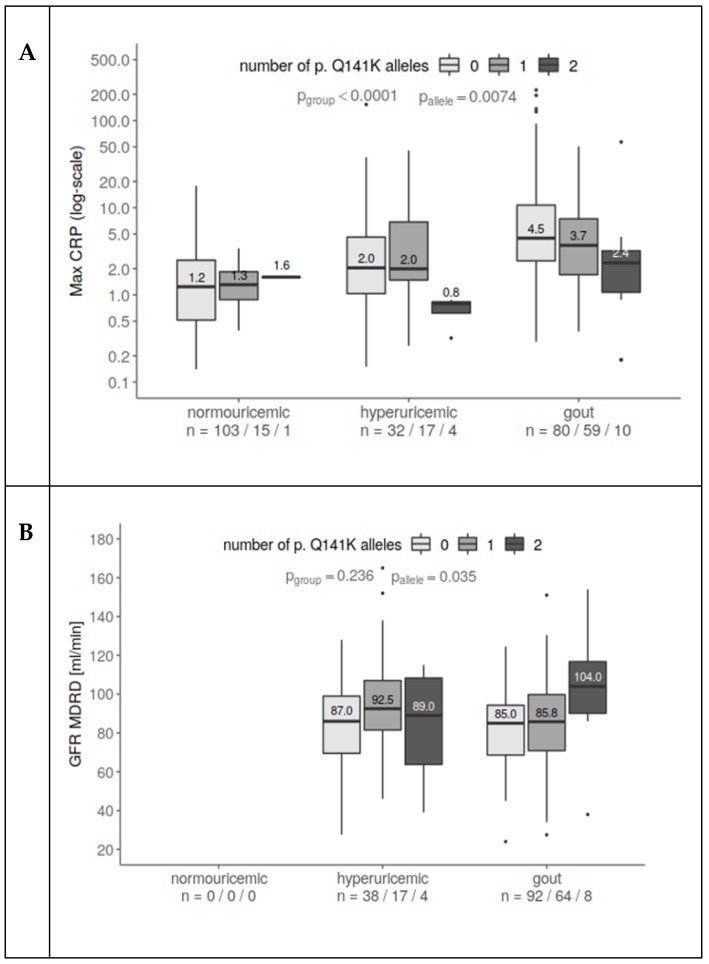
The connection between the presence of p.Q141K, maximum CRP, and eGFR-MDRD. (**A**) CRP increased with hyperuricemia and even more for gout patients compared to normouricemic subjects. At the same time, p.Q141K homozygotes had lower CRP than heterozygotes and wild type. The relationship prevailed after including age as a covariate (CRP slightly rises with subject age). (**B**) GFR increases with an increasing number of p.Q141K alleles, in a similar way for both hyperuricemic and gout patients. One extreme observation (426 mL/min) was excluded since it heavily influenced the fit. All values were log-transformed for a better fit.

**Figure 3 jcm-08-01965-f003:**
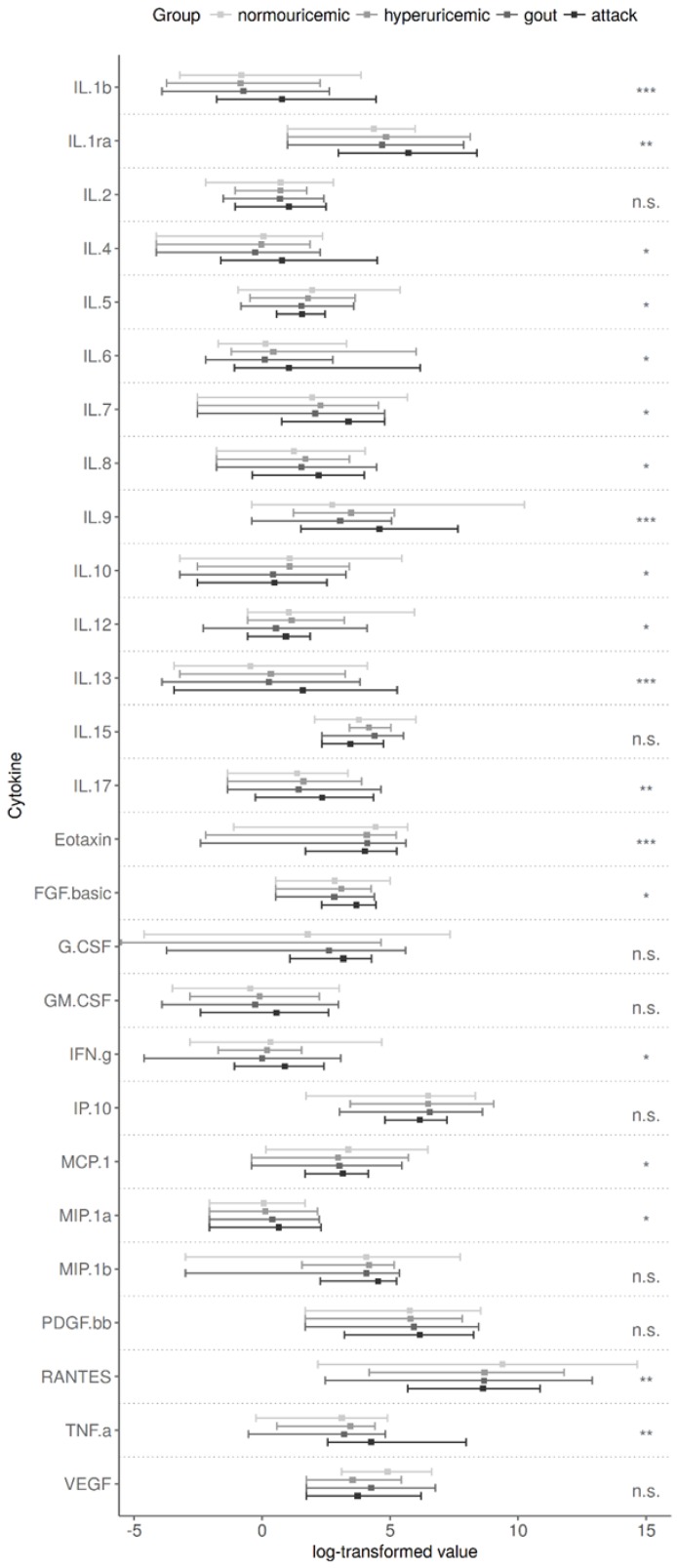
Individual cytokine measurements in the compared groups. Post-hoc pairwise comparisons revealed the main source of differences in the gout attack group were: interleukin (IL)-1b, IL-1ra, IL-4, IL-6, IL-7, IL-8, IL-9, IL-13, IL-17, fibroblast growth factor (FGF)-basic, interferon gamma (IFNγ), and tumor necrosis factor alpha (TNFα). IL-5, IL-10, and IL-12 levels gradually decreased from control to gout attack group. Eotaxin, monocyte chemoattractant protein 1 (MCP-1), and regulated on activation, normal T cell expressed and secreted (RANTES) showed significantly higher values for controls. There were no associations between the p.Q141K allele and cytokine levels. *p*-values via an ANOVA model using log-transformed measurements (except eotaxin) are corrected for multiple comparisons (Benjamini–Hochberg method). n.s. *p* > 0.05, * *p* ≤ 0.05, ** *p* < 0.01, *** *p* < 0.001.

**Table 1 jcm-08-01965-t001:** Main demographic, biochemical, and genetic characteristics of the normouricemic (*n* = 132), hyperuricemic (*n* = 70), and gout-suffering subjects (*n* = 182).

	**Normouricemic Subjects (N = 132)**		**Hyperuricemic Patients (N = 69)**		**Gout Patients (N = 177)**		**Fisher Test *p*-Value**
	*n*	%	*n*	%	*n*	%	
Sex M/F	54/78	40.9/59.1	50/20	71.4/28.6	166/16	91.2/8.8	<0.0001
Familial occurrence	Not collected		31	50.0	66	63.5	0.0717
No treatment	132	100.0	31	44.3	28	15.4	<0.0001
Allopurinol treatment	Not applicable		39	55.7	137	75.3	
Febuxostat treatment			0	0.0	17	9.3	
p.Q141K, GG	103	86.6/78.0	44	64.7/62.9	100	57.5/56.5	<0.0001
GT	15	12.6/11.4	19	27.9/27.1	66	37.9/37.3	
TT	1	0.8/0.8	5	7.4/7.1	8	4.6/4.5	
no data	13	9.8	2	2.9	2	1.7	
p.Q141K, MAF	17	7.1	29	21.3	90	25	<0.0001
Data on interleukins subjects/measurements	132/132		44/53		132/149/23 during attack		
	**Median (IQR)**	**Range**	**Median (IQR)**	**Range**	**Median (IQR)**	**Range**	**KW Test *p*-Value**
Age of onset, years	Not applicable		28.5 (39.2)	1.2–76	42.0 (24.0)	11–84	0.0035
Age now, years	41.0 (25.0)	18–76	38.0 (42.0)	3–78	54.0 (21.0)	11–90	<0.0001
BMI (N = 127/59/146) #	25.5 (4.9)	17.9–38.5	28.1 (6.5)	16–41	28.4 (5.4)	19.5–50	<0.0001
SUA off treatment, µmol/L (N = 132/46/112) #	337 (118.2)	140–617	448 (116.8)	253–608	462 (124.5)	181–683	<0.0001
SUA on treatment, µmol/L (N = 0/42/155) #	Not applicable		424 (140)	240–628	372 (126.0)	163–725	0.0352
FE-UA *	Not collected		3.7 (2.0)	1.6–20	3.6 (1.6)	0.8–14.3	0.6959
eGFR-MDRD, mL/min/1.73 m^2^ *	Not collected		88.0 (36.0)	27.6–426	86.0 (26.0)	24–154	0.2965
Serum creatinine, µmol/L *	75.5 (22.2)	49–121	79.0 (20.5)	26–132	82.0 (20.5)	47–226	0.0016
Max CRP ** (N = 132/54/146) #	1.3 (1.9)	0.1–17.9	1.9 (4.6)	0.2–153.1	4.0 (6.4)	0.2–222.4	<0.0001

* Mean of measurements taken during subject follow-ups. ** Maximum of measurements taken during subject follow-ups. #Some parameters had missing data; in case missing data amounted for 5% or more, real N is mentioned in parentheses. Reference levels: SUA 120–416 µmol/L for men, 120–360 µmol/L for women; FE-UA 7.3 ± 1.3 for men, 10.3 ± 4.2 for women; eGFR-MDRD > 90 mL/min/1.73 m^2^ for healthy subjects (levels decline with age); serum creatinine 64–104 µmol/L for men, 49–90 µmol/L for women; CRP 0–5 mg/L. GG—variant; GT—heterozygotic; TT—homozygotic; MAF—minor frequency allele; IQR—interquartile range; BMI—body mass index; SUA—serum uric acid; FE-UA—fractional excretion of uric acid; eGFR-MDRD—estimated glomerular filtration rate calculated using the Modification of Diet in Renal Disease; CRP—C-reactive protein; KW test—Kruskal–Wallis test.

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
