# Peer review of "Interaction of the p.Q141K Variant of the ABCG2 Gene with Clinical Data and Cytokine Levels in Primary Hyperuricemia and Gout"

_jcm, 2019, doi:10.3390/jcm8111965_

Round 1
Reviewer 1 Report
This is a nicely done study about a mutation in ABCG2 and its possible consequences. This study will add valuable information about these mutations and their occurrence in patients. I cannot fault it. The only criticism I would have is that the figures are borderline small so that not everything is really visible at glance.
Author Response
Thank you for your letter and for the positive comments. We enlarged the figures in the whole article for a better visibility.
Reviewer 2 Report
Proposed paper is well written and interesting, however some minor issue are needed before paper can be accepted.
1- The cardiovascular importance of uric acid is of increasing interest in the last years. Although this is not the focus of this paper it should be encountered in the introduction or in the discussion. In fact the hypotesys for CV damage of uric acid is more related to oxidative stress and to it's xantine oxidase production more than hypoexcretion. Please provide some sentences on this point (please see J Clin Hypertens (Greenwich). 2018 Jan;20(1):193-200 and J Hypertens. 2019 Feb;37(2):380-388).
2- Control population need to be better described in the methods section. Which subjects they are? healthy, hypertensive or what else?
Author Response
1- We thank you for this valuable comment. We have read the suggested articles with interest but unfortunately, we are not able to comply with your recommendation for citations. The proposed papers are not related to the topic of our research and we believe that adding the information would jeopardize the clarity of our article. We appreciate the suggestions and the importance of uric acid in CV damage, but our study focuses solely on the genetic predisposition to hyperuricemia and gout. It would obviously be very valuable to statistically analyze the relationship of p.Q141K variant with CV damage in a future study.
2- Thank you for your comment. We tried to be more specific and we added phrase „normouricemic individuals from general population with no history of hyperuricemia, gout or autoimmune disease“. The control group was selected from the personnel of the Institute of Rheumatology. They were healthy individuals without known metabolic syndrome, although they were slightly overweight (median BMI was 25,5) and they had no history of chronic inflammation. This was documented not only by their medical history but also by the low level of inflammatory biomarkers (median CRP was 1.3) We hope that now it is clearer.
Reviewer 3 Report
This is manuscript addresses clinical immunological parameters of numerous patients carrying the ABCG2 Q141 variant, which has been linked with the development of gout but also multidrug resistance in cancer. The authors provide an interesting correlation between the presence of the Q141 allele and the age-dependent onset of clinical gout phenotypes, especially in patients with autoimmune disorders such as acute gouty arthritis.
Overall, this is an interesting contribution on the role of Q141 in the onset of gout. The study is well-done and the statistics is satisfactory.
Author Response
We appreciate your positive comment. Thank you very much for reading and evaluation of our study.